# The Role of Integrins for Mediating Nanodrugs to Improve Performance in Tumor Diagnosis and Treatment

**DOI:** 10.3390/nano13111721

**Published:** 2023-05-24

**Authors:** Chi Yu, Wei Jiang, Bin Li, Yong Hu, Dan Liu

**Affiliations:** 1College of Pharmaceutical and Biological Engineering, Shenyang University of Chemical Technology, Shenyang 110142, China; 2Institute of Materials Engineering, College of Engineering and Applied Sciences, Nanjing University, Nanjing 210093, China; 3Department of Biochemistry and Molecular Biology, Medical College, Guangxi University of Science and Technology, Liuzhou 545005, China

**Keywords:** integrin, nanodrugs, drug delivery, tumor diagnosis, tumor treatment

## Abstract

Integrins are heterodimeric transmembrane proteins that mediate adhesive connections between cells and their surroundings, including surrounding cells and the extracellular matrix (ECM). They modulate tissue mechanics and regulate intracellular signaling, including cell generation, survival, proliferation, and differentiation, and the up-regulation of integrins in tumor cells has been confirmed to be associated with tumor development, invasion, angiogenesis, metastasis, and therapeutic resistance. Thus, integrins are expected to be an effective target to improve the efficacy of tumor therapy. A variety of integrin-targeting nanodrugs have been developed to improve the distribution and penetration of drugs in tumors, thereby, improving the efficiency of clinical tumor diagnosis and treatment. Herein, we focus on these innovative drug delivery systems and reveal the improved efficacy of integrin-targeting methods in tumor therapy, hoping to provide prospective guidance for the diagnosis and treatment of integrin-targeting tumors.

## 1. Introduction

Tumors, as one of the most high-incidence diseases that seriously threatens human health, have become a leading cause of death worldwide [1]. Frequent invasion and metastasis have become important biological characteristics of malignant tumors as well as the main factors affecting the clinical therapeutic effect and prognosis of tumors [2,3]. Recently, it has been discovered that several proteins, including adhesion receptor families, receptor tyrosine kinases, cytoskeleton proteins, and adaptor and signaling proteins, are linked to the invasion and metastatic properties of tumor cells [4,5,6]. Among them, integrins, as a member of adhesion receptor families, comprise a wide number of cellular receptors and are important proteins on the cell surface [7,8]. They are involved in practically every stage of tumor progression, from initial tumor formation to metastasis [9].

Integrins are transmembrane glycoproteins that attach the cells to the extracellular matrix (ECM) and serve as bidirectional hubs transmitting signals to mediate cell-cell and cell-ECM interactions [10,11,12]. They modulate tissue mechanics and regulate intracellular signaling, including cell faith, survival, proliferation, and differentiation, and the up-regulation of integrins in tumor cells has been confirmed to be associated with tumor development, invasion, angiogenesis, metastasis, and therapeutic resistance. Thus, reasonably regulating the activity of integrins in tumor cells can effectively improve the therapeutic effect of oncotherapy and inhibit tumor invasion and metastasis.

Recently, with the rapid development of nanotechnology, nanodrugs have attracted wide attention due to their excellent biocompatibility and modifiability. As an effective target for tumor therapy, integrins can not only be used to regulate tumor microenvironment but also can promote nanodrugs accumulation in tumor tissues. Herein, we will review the structural characteristics, biological functions, mechanisms of action of integrins in tumor cells, the research progress of the interaction between integrins and tumor microenvironment, as well as the application of integrin-targeting nanomedicine in tumor therapy.

## 2. The Structure and Function of Integrins

### 2.1. The Structure of Integrins

Integrins, as adhesion molecules and transmembrane receptors that consist of α and β subunits, heterodimers, are widely located on the surface of vertebrate cells and are involved in cell-cell and cell-matrix interactions [11]. Integrins feature an extracellular domain, a single-pass transmembrane domain and a short cytoplasmic domain (except β4) [13]. Of these, the extracellular domain can interact with ECM ligands to transmit signals [13,14]. Generally, there are 18α and 8β subunits that can combine to generate 24 distinct receptors with varying binding capabilities and tissue distribution [15,16]. Specifically, nine of these α subunits contain an αI domain at the N-terminus consisting of approximately 190–200 amino acid residues and can be strongly bound by ligands [17]. Moreover, metal ion binding sites at the top of the αI domain have been demonstrated to bind divalent metal ions, such as magnesium ions and manganese ions, thus, mediating the interaction of integrins and ligands [18,19]. For the α subunits that lack the I domain (α3, α4, α5, α6, α7, α8, α9, αv, and αIIb), the β-propeller of the α subunits and the I domain of the β subunit can be joined to form the binding site [20]. Moreover, many integrin ligands, as well as the integrins themselves, present structural alterations under specific conditions, which may compromise signaling pathways that are constitutively triggered under physiological conditions [21,22].

Of the 24 human integrin subtypes known to date, eight integrin dimers can recognize Arg-Gly-Asp (RGD) peptide in extracellular matrix proteins [23]. In addition, there are four types of integrins based on their connective ligand: leukocyte, collagen-binding, Arg-Gly-Asp (RGD)-binding, and laminin-binding integrins (Figure 1). The short cytoplasmic domain provides binding sites for these adaptors, signaling proteins, and cytoskeleton-associated proteins, all of which are required for bidirectional integrin signaling [12,24]. Moreover, the integrins also serve as receptors for certain toxins (such as snake venom), viruses, and other pathogens [25]. In addition, fibronectin is regarded as a potential integrin ligand and the extra deposition of fibronectin in tissue leads to inflamed diseases [26,27].

### 2.2. The Function of Integrins

Integrins are essential for cell growth and survival. They tightly link cells to ECM and deliver signals both internally and outwardly, acting as a communication bridge between the internal and external worlds of cells [28]. Integrins are activated in the extracellular domain via conformational rearrangement from bent-closed to open-active states [29,30]. In this process, the ECM ligand binding would rapidly lead to the conversion of the integrins to the high-affinity extended-open form that permits coupling to the actin cytoskeleton. This bridging to actin supports cell adhesion and the recruitment of additional intracellular binding partners that mechanically reinforce the link and allow subsequent downstream signal transduction [31]. The bidirectional transmembrane signals are delivered through “inside-out” and “outside-in” pathways (Figure 2). Briefly, some intracellular signal molecules, such as talin and paxillin, can bind to the intracellular region of integrins, inducing conformational changes in the intracellular region and initiating a series of subsequent conformational changes in the extracellular domain, a process known as “inside-out” signaling [32]. In this process, the binding of ligands to integrins causes changes in the intracellular cytoskeleton and the recruitment of important intracellular signal molecules [33]. Furthermore, the intracellular domains of α and β subunits are separated and bind to intracellular signal molecules, activating relevant intracellular signaling pathways and mediating cell functions. This is known as “outside-in” signal transduction. The bidirectional signal transduction ability of integrins is critical for cell proliferation, invasion, angiogenesis, and apoptosis resistance, contributing to more aggressive diseases, such as tumors [8,34]. For example, Integrin α2 (ITGA2) is overexpressed in solid tumors, including pancreatic, gastric, liver, prostate, and breast tumor, and its expression level is highly correlated with tumor proliferation and invasion [35,36,37,38]. In addition, integrin β1, β3, β6, αv, and α5 are also found in many tumors and play an important role in tumor therapies [39,40,41]. Therefore, an in-depth understanding of the interaction between integrins and tumors is beneficial in improving tumor therapy.

### 2.3. The Effect of Integrins on Tumors

Integrins have been found to play multiple key functions in practically every stage of tumor progression, from basic tumor growth to metastasis, and hence integrin modulation has been widely used in tumor therapy [9,43,44]. For example, the pre-treatment of cell monolayers with different matrix components (laminin, fibronectin, and collagen), makes neoplastic cells more resistant to cytotoxic insults mediated by different classes of chemotherapeutic agents in an event dependent on integrin-ECM components interaction [45]. The overexpression of ECM components was proven to result in chemoresistance [46,47,48]. However, because tumor incidence and growth are exceedingly complicated, many concerns remain unsolved, particularly the impact on tumor metastasis and angiogenesis [49,50,51]. Further research into the mechanisms of the integrins family in tumor metastasis and tumor angiogenesis will help to expand the use of integrins in tumor therapy.

Compared with normal tissues or cells, the expression of integrins is maintained at a high level in tumors and plays an important role in tumor proliferation and invasion [52,53,54]. For example, integrin β1, a member of the integrins family, has been confirmed to be overexpressed in nasopharyngeal carcinoma tissues [37]. It promotes the migration and metastasis of nasopharyngeal carcinoma cells and reduces the sensitivity of cells to radiotherapy [55]. It has been discovered that the inhibition of integrin β1 expression can effectively improve the radiation sensitivity of nasopharyngeal carcinoma [56]. In addition, the high expression of integrin α5β1 in hepatocellular carcinoma is closely associated with the occurrence and metastasis of hepatocellular carcinoma, the antagonists of integrin α5β1 can significantly reduce the migration ability of hepatocellular carcinoma [57,58].

## 3. The Role of Integrin αvβ3 in Nanodrugs Antitumor Therapy

Nanotechnology has a wide range of applications in medicine, such as nanodrugs delivery, nano contrast agents and medical materials. Compared with small molecule drugs, nanodrugs have the characteristics of size effect, large specific surface area, and photothermal effect, which helps to prolong the biological half-life of drugs in vivo, and improve the antitumor effect and imaging ability [59,60,61]. Previous studies have shown that integrin αvβ3 expression on the surface of tumor cells is substantially higher than in normal tissues and that it is easily recognized and bound by RGD and other molecules [27,54,62] (Figure 3). Therefore, the design of nanodrugs targeting integrin αvβ3 is of great practical significance for tumor treatment and diagnosis by enhancing the targeting of nanodrugs in tumor tissues, achieving optimal antitumor potency with minimized toxicity.

### 3.1. Chemotherapy

Chemotherapy is a common form of clinical antitumor therapy. Although chemotherapy has shown some efficacy in antitumor treatment, drug enrichment in tumors is limited due to the in vivo complex environment and physiological obstacles [63,64,65]. According to the report from Warren Chan et al., the amount of nanodrugs entering tumor tissue is even less than 1% of the injected amount [66]. Although this conclusion is controversial, it is not satisfactory to show that nanodrugs only relies on the enhanced permeability and retention (EPR) effect to enter tumor tissue and exert antitumor effects [67]. Therefore, some studies use antibodies and other means to analyze drug aggregation at the tumor site, hoping to improve the low-targeting efficiency [68,69,70]. Currently, chemotherapy based on targeting integrins has attracted widespread attention, mainly for better drug targeting, reducing the off-target effect, and improving the antitumor efficacy [71,72]. The iRGD is a bifocal ligand containing RGD and a C-end sequence (CendR), which can target integrin αvβ3 and neuropilin-1, respectively [73]. For instance, polylactic acid-glycolic acid copolymer (PLGA) was modified by iRGD to form iRGD-PLGA-PTX nanomicelles. The nanomicelles were then intravenously injected into colorectal tumor-loaded mice. The results showed that the inhibition rate of the iRGD-PLGA-PTX group was higher than the PLGA-PTX group, indicating that iRGD promoted endocytosis of PLGA-PTX nanomicelles by colon tumor cells and improved the antitumor efficiency (Figure 4) [74]. In another instance, PLGA combined with RGDfk to prepare nanodrugs for the treatment of breast tumor. PLGA is often used in drug and gene delivery due to its excellent biocompatibility, high stability, and degradability [75,76]. The RGDfk-modified nanodrugs can not only increase drug solubility but also improved the biological activity of the drug and prolonged the half-life of the drug in vivo [77]. In addition, the EGF-Like Repeats and Discoidin I-Like Domains Protein 3 (EDIL3) was highly expressed in paclitaxel-resistant breast tumor, but low in sensitive breast tumor. Ding, Y et al. revealed that blocking the interaction between integrin αvβ3 and EDIL3 by Cilengitide improved the sensitivity of tumor tissues to paclitaxel and reversed the epithelial-mesenchymal transition (EMT) of paclitaxel-resistant tumors, reducing the probability of tumor resistance [78].

### 3.2. Radiotherapy

Radiotherapy can induce the instantaneous, dynamic, and local “bursting” of tumor microvessels, thus, increasing the permeability of blood vessels and facilitating the entry of nanoparticles from outside the blood vessels into the tumor, promoting drug enrichment in tumor site, enhancing the local drug concentration, and reducing drug resistance [79,80]. Gold nanoparticles are often regarded as radiosensitizers because of their unique physical and chemical properties and excellent biocompatibility [81]. Gold nanoparticles can kill tumor cells at low doses of radiation, reducing the damage of radiation therapy to normal tissues [82]. For in vivo CT imaging and radiosensitization evaluation, c(RGDyC)-AuNPs enhanced the sensitivity of radiotherapy and improved the effect of radiotherapy in 4T1 tumor-bearing mice with integrin αvβ3 modified nanodrugs [83]. Compared with gold, copper (64) is also a radiosensitizer. The ^64^Cu-Pyro-3PRGD_2_ probe was synthesized by 3PRGD_2_-modified Pyropheophorbiide-A (Pyro), which showed high-tumor specificity in both positron emission tomography (PET) and optical imaging [84]. 3PRGD_2_ has a higher affinity with integrin αvβ3, which can reduce drug dose and tumor resistance, and inhibit tumor metastasis [85]. Meantime, 64Cu-Pyro-3PRGD_2_ has good water solubility and has very low accumulation in normal organs, which can be quickly cleared through renal metabolism to avoid potential damage to normal tissues [84]. Platinum drugs are not only one of the chemotherapeutics but also used in clinical radiotherapy sensitizers [86]. By modifying platinum-based nanoparticles (PtNPs) with RGD to form RPNs, the uptake of cisplatin by tumor cells was enhanced through RGD targeting, and DNA fragmentation was promoted [76]. In vitro and in vivo experimental results demonstrated that the RPNs + X-ray group had a higher level of tumor apoptosis and lower survival rate of nasopharyngeal carcinoma cells than the RPNs alone treatment group, confirming that RPNs combined with X-ray enhanced the inhibitory effect on the nasopharyngeal carcinoma cells (Figure 5).

In addition, targeted radionuclide therapy (TRT) is also an important component of radiotherapy [87]. By being encapsulated into nanocarriers, radionuclides can naturally accumulate in tumors by positive targeting [88,89]. The TRT radiotherapy can be used for local radiotherapy and metastatic tumor. For example, the programmed death ligand 1 (PD-L1) combined with RGD-modified TRT (^177^Lu as a radionuclide) nanoparticles was used in colon tumor-bearing mice [90]. The CT imaging studies in small animals have demonstrated that anti-PD-L1 antibody combined with integrin αvβ3-targeted ^177^Lu-EB-RGD nanodrugs can improve antitumor efficacy with reduced tumor resistance, prolonging overall survival compared with other groups [90].

### 3.3. Immunotherapy

Immunotherapy is a rapidly developing antitumor strategy in recent years with high clinical application value [91]. The integrin-based targeting delivery system can transport the radiosensitizer GTe (L-glutathione-decorated triangle-shaped nanodrugs) to tumor cells, enhancing the X-ray sensitivity of tumor cells, inducing reactive oxygen species (ROS) to destroy mitochondria and promote cell apoptosis [92]. Compared with GTE-RGD + X-ray therapy, GTe-RGD + X-ray/anti-PD-1 has a more significant antitumor effect, suggesting that GTE-RGD + X-ray/anti-PD-1 can activate immune function and inhibit tumor growth. 1-methyl tryptophan (1MT) inhibits indoleamine 2, 3-dioxygenase (IDO) specifically [93]. Integrin-targeting DOX@MSN-SS-iRGD&1MT (iRGD-modified doxorubicin and 1MT co-loaded nanodrugs) were synthesized as previously reported [94]. In recent research, an integrin-facilitated lysosomal degradation molecular was designed by a cyclic RGD peptide connecting with PD-L1 [95]. This method allowed integrin-facilitated lysosomal degradation of PD-L1 by bifunctional compound to enhance antitumor immunotherapy. In other researches, the inhibition of integrin αvβ6 and αvβ3 also showed a remarkable antitumor effect in different tumors [96,97,98]. The results showed that iRGD-modified nanoparticles were enriched in tumor sites, inhibiting the tumor proliferation and also up-regulating the differentiation of cytotoxic CD8^+^ T cells, confirming that the integrin-targeting nanoparticles can activate the immune response against tumor (Figure 6).

### 3.4. Other Treatment Strategies

Photodynamic therapy (PDT) has attracted increasing attention due to its advantages of low damage to normal tissues, low invasion, economy, and little drug resistance [99,100]. However, the effect of PDT therapy is limited by the accumulation of photosensitizer (PS), poor tumor targeting, and hypoxic tumor microenvironment, so it is worth studying to improve PDT by improving photosensitizer accumulation in tumors [101,102]. Au-RGD-miR-320a nanoparticles targeting integrin αvβ3 were constructed for enhanced photosensitive therapy and gene-targeted therapy. This nanodrugs combined with laser irradiation can significantly inhibit the proliferation and metastasis of lung tumors and enhance the apoptosis of tumor cells, and can effectively prevent tumor metastasis. In addition to optimizing PS delivery, the combination of laser irradiation also improves drug release behavior of other antitumor drugs [103,104]. Laser irradiation can also be used for nanocarriers to improve oxygen content in tumor tissues, alleviating tumor hypoxia and slowing down tumor drug resistance. Avraham Dayan’s lab has developed an integrin-targeting nanophotodynamical drug [105]. By modifying TiO_2_ with the RGD group, a nanobiological complex is generated, and when combined with laser irradiation, can specifically bind to tumor cells, promoting sufficient drug release and cell death, thus, improving the therapeutic effect and reducing drug resistance (Figure 7). Combining photothermal therapy with integrin-targeting nanodrugs can not only reduce the damage to normal tissues but also improve the antitumor effect, making this a promising tumor treatment strategy [106].

## 4. The Nanoparticles Targeting Integrin αvβ3

Tumor-targeting therapy is the administration of antitumor drugs to tumor tissue to prevent tumor formation, migration, and invasion [107]. Compared with traditional chemotherapy and surgery, tumor-targeted therapy has a higher tumor suppressive effect with reduced side effects [108,109].

### 4.1. Protein-Based Nano-Delivery System

Albumin is currently used in clinical antitumor therapy due to its excellent physiological properties when combined with paclitaxel to form nanodrugs [110]. RGD polypeptide targeting integrin αvβ3 was modified with ^131^I-loaded bovine serum albumin (BSA) to construct RGD-BSA-polycaprolactone (^131^I-RGD-BSA-PCL) [111]. It was found that although both BSA-PCL with or without RGD could produce fluorescence after being co-cultured with cells, the tumor absorption rate of BSA-PCL with RGD NPs was higher than that of BSA-PCL without RGD NPs, indicating that integrin αvβ3 could be a molecular target in tumor diagnosis and treatment.

### 4.2. Polymers

Pegylated paclitaxel has been widely used in clinical antitumor therapy since it was approved by FDA [112]. This is mainly because polyethylene glycol (PEG) has low toxicity, high biocompatibility, and negligible immunogenicity [113,114]. Therefore, studies of various polymer micelles for nanodrugs delivery are still attracting a lot of attention. Amphiphilic nanopolymers, which contain both hydrophilic and hydrophobic chains, can self-assemble to form nanoparticles in an aqueous solution [115]. The hydrophilic part is located on the surface while the hydrophobic chain and antitumor drugs are wrapped in the inner side, allowing the antitumor drugs to reach the tumor tissue through the circulatory system [116,117]. To improve the target efficacy of the polymer, integrin αvβ3-based targeting nanomicelles were constructed for increased drug concentration and improved tumor death [118]. At the same dose of doxorubicin (DOX), the RGD-modified polymer micelles inhibited the growth of tumor cells more significantly than free doxorubicin [119]. More importantly, there was no obvious toxic side effect in the RGD-modified polymer micelles group in normal organs.

### 4.3. Inorganic Nanomaterials

In recent years, inorganic nanoparticles have attracted much attention in tumor imaging, photothermal therapy and radiosensitization therapy, including copper sulfide nanoparticles, magnetic nanoparticles, mesoporous silica nanoparticles, gold nanoparticles, and quantum dots [120,121,122]. Integrin-based strategies were implemented as an effective means of achieving active tumor targeting to further improve the antitumor effect of nanodrugs. Targeting molecules are usually modified with inorganic nanomaterials to enable nanodrugs to aggregate in tumor. For example, RGD-modified gold nanoparticles (^125^I-RGD@AuNPs) were constructed to promote the specific targeting of gold nanoparticles by integrin αvβ3 receptor [123]. In tumor tissues, γ rays were released from ^125^I-RGD@AuNPs to inhibit tumor cell growth. RGD-modified polyamide dendrimers coated with gold nanoparticles were also designed for in vivo imaging, improving the biocompatibility of gold nanoparticles [124]. The CT imaging effect of the nanomaterial is superior to that of the clinical contrast agent iodihydramol (Omnipaque). At low-dose X-ray, RGD-modified gold nanoparticles showed a higher signal-to-noise ratio and more obvious lung MR and CT imaging effects than iodihydramol. In addition, targeting nanoprobe cRGD-IONPs were constructed based on ultrafine magnetic iron oxide nanoparticles (IONPs). The ^125^I labeled cRGD-IONPs probe can be enriched in tumor sites by cRGD targeting with integrin αvβ3 receptor, showing excellent tumor imaging [125]. Quantum dots (QDs) are nanomaterials with unique fluorescence properties that have promising applications in tumor imaging and early detection [126,127]. The QDs-RGD probe coupled with RGD peptide showed an obvious tumor imaging effect in SW1990 pancreatic tumor model [128]. When QDs-RGD NPs were combined with gemcitabine, the tumor inhibition rate of the combined treatment group (QDs-RGD NPs+ laser irradiation + gemcitabine) was much higher than that of the gemcitabine group alone, suggesting that QDs-RGD NPs has a synergistic effect on inhibiting the proliferation of SW1990 pancreatic tumors.

### 4.4. Membrane

Recently, a new direction of research is decorating nanoparticles with cell membranes to prevent drugs from being captured and removed by the immune system [129]. Among them, red blood cells, stem cells, platelets, phagocytes, and tumor cells are effective natural carriers with excellent biocompatibility and a unique ability to evade immune system [130,131,132,133]. For example, nanoparticles modified by homologous hepatocellular carcinoma cell membranes or neutrophil membranes can improve nanodrugs retention in tumor sites and effectively inhibit tumor growth [134,135]. An RGD-decorated macrophage membrane was coated on the magnetic nanocluster to deliver siRNA. Such feature-packed magnetic nanodrugs enable us to gain success in high-performance siRNA delivery through superior stealth effect, magnetic resonance imaging, magnetic accumulation, RGD targeting, and favorable cytoplasm trafficking [136]. Similarly, RGD peptide-modified platelet membranes coated with melanin nanoparticles and doxorubicin (DOX) have a significant inhibition effect on the proliferation and metastasis of drug-resistant tumors [137].

## 5. Physical Barriers to Integrin-Targeting Nanodrugs

Integrins are overexpressed on the surface of tumor cells [52,54]. The aggressive growth of tumors requires a lot of nutrients to promote the formation of new blood vessels [138]. Endogenous αvβ3 integrin transmembrane glycoprotein, which is widely distributed in tumor neovascularization, is thought to be an angiogenesis marker [8,139]. Integrins and immunoglobulin superfamily cell adhesion molecules (IgCAM) and junction adhesion molecules (JAM) have a series of interactions, including leukocyte extravasation in the blood, immune monitoring in the gut, and hematopoietic stem cells homing and mobilization [140]. In addition, ECM ligands enhance interactions between pathogens and phagocytic immune cells, acting as phagocytic primers and inducers for neutrophil extracellular traps [141]. Integrins can also activate signal transduction pathways after binding to ECM, mediating cell signals in cell growth, survival, division, and migration [142]. Therefore, integrins were often used for specific ligands for targeting drug delivery (targeting tumor cells or ECM). Integrins can be also used to reduce the loss of nanodrugs and increase their half-life and bioavailability in vivo [143,144].

### 5.1. Opsonization

Opsonization is the molecular mechanism whereby molecules, microbes, or apoptotic cells are chemically modified to have stronger interactions with cell surface receptors on phagocytes and antibodies to identify invading particles. Integrin-targeting nanodrugs need to reduce opsonic as much as possible in vivo to avoid phagocytosis by macrophages [145]. Due to the surface properties, many serum proteins (also called opsonins) can be deposited on the surface of nanoparticles or can enter the nanoparticles to form dynamic protein crowns (the protein-rich layer formed around nanoparticles), which can be effectively recognized by the immune system and assimilated by macrophages [146,147]. Integrin modification can increase the targeting efficiency of nanodrugs and reduce the excessive adhesion of nanodrugs to cells, thus, enhancing the efficacy of nanodrugs [148].

As is known to all, liposomes have higher drug delivery efficiency and wider drug delivery spectrum than small molecules and can achieve targeting drug delivery after being combined with targeting units [149,150,151]. The protein crown formed by liposomes and plasma proteins after entering blood circulation has a certain influence on the targeting function of liposomes [152,153]. Protein crowns of liposomes are rich in opsonins and coagulation proteins can activate immune cells and promote phagocytes to absorb nanoparticles through receptor-mediated phagocytosis [154,155] In Fang et al.’s study, PEG-PCL was cross-linked with RGD and coated with DOX to form DOX-cRGD-RCCMS nanoparticles, and the metabolic effects of the cross-linked nanodrugs showed an elimination half-life of 4.7 h, 11.7 times longer than the free DOX (0.4 h), and 3.9 times longer than the non-cross-linked DOX-cRGD-PEG-PCL (1.2 h), demonstrating the important role of RGD cross-linked nanomicelles in drug pharmacokinetics in vivo [156]. Therefore, RGD crosslinked nanodrugs can prolong their half-life in plasma and reduce their opsonance (Figure 8).

### 5.2. Penetration in Tumor

The combination of nanoparticles and targeted integrins increases the targeting efficiency of nanodrugs. However, the complex tumor microenvironment possesses unique structure and physiological functions, which not only hinders tumor penetration of chemotherapeutics but also becomes a physiological barrier to tumor penetration of integrin-targeting nanodrugs [157]. As a result, many antitumor drugs can only be enriched on the surface of tumor tissues rather than entering the deep tumor tissues and inhibiting tumor proliferation or metastasis [158]. iRGD is a tumor-targeting transmembrane peptide of circular RGD peptide linked by a disulfide bond, which can bind specifically to integrins and mediate the penetration of tumor cells [159]. The iRGD-heparin nanocarrier iRGD-Heparin (iHP) modified by iRGD has confirmed that iHP can target tumor cells for cisplatin and dye delivery, which can provide accurate diagnosis and treatment for gastric tumor and reduce the toxicity of drugs in normal tissues. To further demonstrate the effect of peptides on cell penetration, the uptake capacity of PEGylated PLGA nanoparticles modified by cRGD at different densities in cells was studied. Confocal microscopy showed that the penetration of nanoparticles into U87MG 3D multicellular spheres was enhanced with the increase incRGD density. The localization and penetration of these nanomaterials to lysosomes were also observed in 2D culture. In addition, the effects of the same incubation time and temperature on the penetrability of cRGD-modified polymer micelles (CCPM) preparations with different densities were compared. The degree of cell internalization increased with the increase in cRGD concentration. Under different culture conditions, CCPM uptake of 5% cRGD was observed to be significantly higher than 0% cRGD by immunofluorescence staining, which proved that integrins receptor-mediated CCPM uptake was enhanced with the increase in surface CRGD [160].

### 5.3. Endocytosis

Various factors in vivo can interfere with nanodrugs, leading to poor drug targeting and difficulty in entering tumor tissues [158]. Thus, most of them enter normal tissues, resulting in the release of the tumor treatment drugs loaded in the nanomaterials into normal tissues, thereby increasing the toxic side effects on the body [59,161]. Nanoparticles wrapped in cell membranes can disguise as autologous cells to avoid elimination by the immune system [162]. For instance, natural neutrophil membranes wrapped on the surface of synthetic nanostructures constitute a potential biomimetic nanoparticle. cRGD and non-targeted control polypeptide cRAD were modified with nanoparticles to explore the mechanism of action of targeted drugs in vivo (Figure 9). The distribution and metabolic effects of cRAD-NPs and cRGD-NPs in mice were compared using living microscope and flow cytometry. The results confirmed that cRGD-NPs were transferred to the tumor using phagocyte hitchhiking, revealing that neutrophils are the main phagocytic contributor to this process. The results also revealed that neutrophils showed a clear preference for cRGD-NPs. Therefore, cRGD-NPs injected into the microbe were widely disseminated into tumor tissues by immune cells [163]. In another research, DOX was loaded in RGD-modified liposomes, the loaded liposomes with or without RGD modification and free DOX were tested for antitumor effect [164]. The results showed that DOX-loaded RGD-liposomes could effectively target and inhibit tumors and significantly improve the survival rate of tumor-bearing mice. At the same time, DOX-loaded RGD-liposomes can reduce the aggregation and release of drugs in normal tissues while abundantly accumulating in tumor site. Therefore, DOX-loaded RGD-liposomes can be used as a safe and effective antitumor drug delivery system to improve antitumor efficacy.

## 6. Conclusions and Perspectives

Integrin-mediated adhesion between cells and cells to the ECM is crucial for the physiological development and functioning of tissues, and targeting integrins has become an effective means to improve the efficacy of tumor therapy. In this review, the structure and function of integrins, the role of integrin αvβ3 in antitumor therapy, the nanoparticles targeting integrin αvβ3, as well as the physical barriers to integrin-targeting nanodrugs are well discussed. We systematically analyzed the advantages and enhancement strategies associated with integrin-targeting nanodrugs, hoping to lay the foundation for the potential future application of tumor diagnosis and treatment with nanodrugs therapies.

The investigation into the relationship between integrin and tumors remains ongoing, with increasing clarity regarding the mechanism of action of integrin in tumor therapy. With the discovery of new integrins or ligands, the role of integrins in tumor treatment will become even more important; other types of integrins, such as ανβ6, α4β1, α2β1, and α5β1 are also gaining interest. More importantly, the development of nanomedicine has led to the emergence of a variety of functional nanomedicines, and integrin is poised to become the preferred choice for targeted delivery due to its impact on tumors.

Tumor-targeting nanodrugs are an emerging area that has begun to show clinical success, and various nano-based diagnostic and therapeutic drugs are currently undergoing clinical trials (ClinicalTrials.gov). Despite the results to date, there are still many challenges and limitations that need to be resolved. Drug carriers should effectively deliver cargo directly to target cells rather than normal organs and tissues. In the meantime, the carriers should possess the ability of biocompatibility and biodegradability, avoiding extra toxicity. Moreover, these carriers must have long circulation to avoid premature drug release. Integrins possess the ability to specifically interact with target cells to enhance drug delivery efficiency and accuracy. However, integrins are also expressed in normal cells; thus, integrin-targeting methods have hidden dangers of off-target effect. This may cause severe side effects in clinical treatments. Overall, we believe that research efforts on integrin-targeting nanodrugs that are used for antitumor therapy will result in highly efficient, safe, and curative treatment modalities that meet the prerequisites of personalized medicine and achieve permanent cures without side effects.

## Figures and Tables

**Figure 1 nanomaterials-13-01721-f001:**
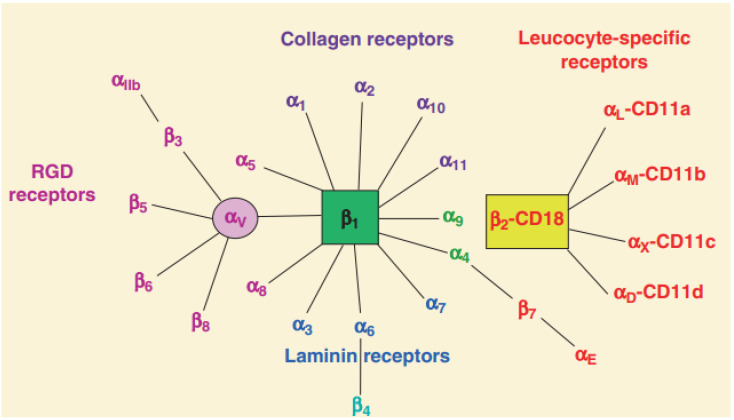
Classification of integrins family [8]. Integrin heterodimers were composed of different combinations of α and β subunits. In terms of ligand specificity, integrins can be divided into four groups: collagen-binding integrins (α1β1, α2β1, α10β1, and α11β1), RGD-recognizing integrins (α5β1, αVβ1, αVβ3, αVβ5, αVβ6, αVβ8, and αIIbβ3), laminin-binding integrins (α3β1, α6β1, α7β1, and α6β4), and leukocyte integrins (αLβ2, αMβ2, αXβ2, and αDβ2). The β2 integrin subunit (CD18) can pair with one of the four α subunits (αL-CD11a, αM-CD11b, αX-CD11c, and αD-CD11d), forming different leukocyte function-associated antigens.

**Figure 2 nanomaterials-13-01721-f002:**
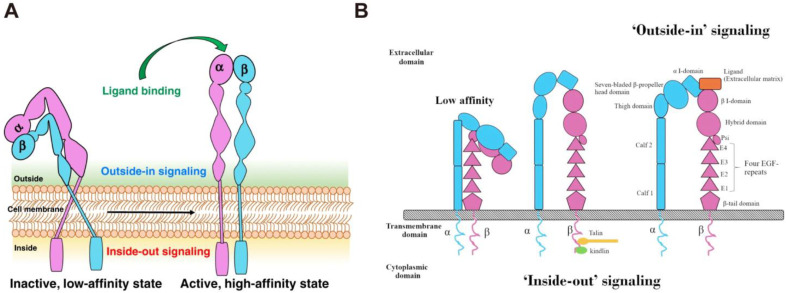
(**A**) The structure and activation pathways of integrins [8]. Integrins in the bent resting conformation reveal low-affinity binding to their ligands. After being activated, integrins were able to have a high affinity with their ligands. (**B**) Schematic illustration of “Outside-in” and “Inside-out” integrins signaling pathways [42]. Integrins can be activated from two directions, from the inside by the regulated binding of proteins to the cytoplasmic tails, and from the outside by multivalent ligand binding. For “Outside-in” signaling, signals received by other receptors foster the binding of talin and kindlin to the cytoplasmic end of the integrin β subunit, at sites of actin polymerization. As for “Outside-in” signaling, ligand binding to the external domain causes conformational changes that increase ligand affinity, modify protein-interaction sites in the cytoplasmic domains and thence the resulting signals.

**Figure 3 nanomaterials-13-01721-f003:**
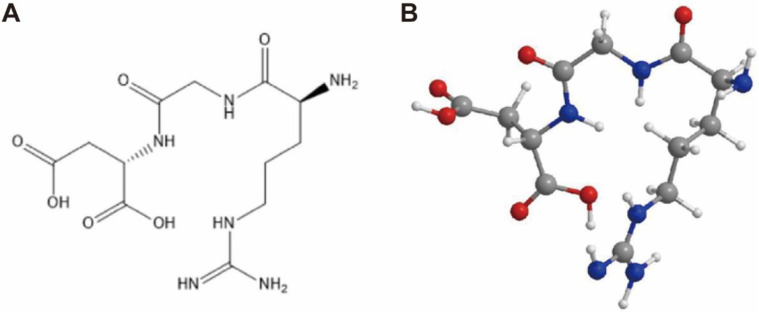
(**A**) Structure diagram of RGD sequence. (**B**) Ball-and-stick model of RGD sequence.

**Figure 4 nanomaterials-13-01721-f004:**
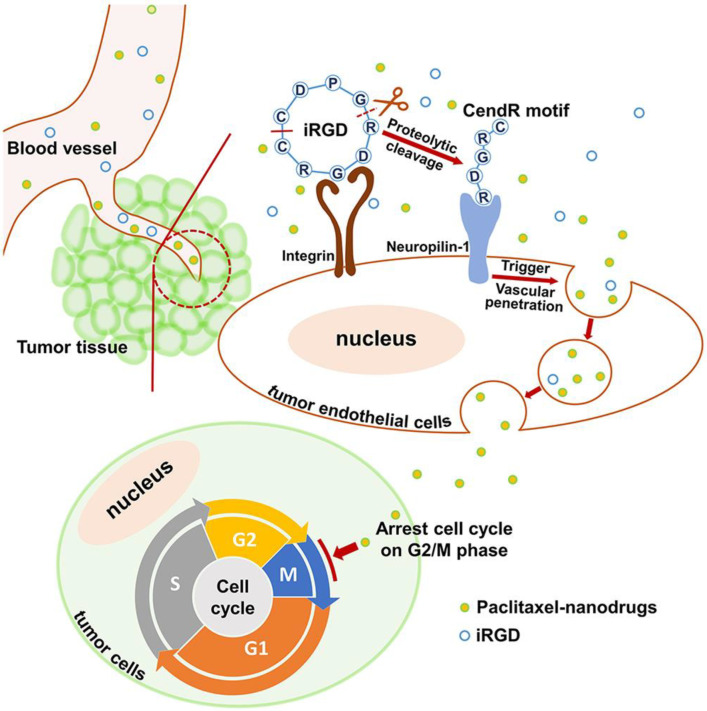
Schematic illustration of iRGD-PLGA-PTX nanomicelles used in antitumor research [74]. When the co-administered nanodrugs and iRGD peptide reach the tumor tissue, iRGD binds to the integrin receptor and is proteolytically cleaved. The exposed tissue-penetrating motif CendR interacts with the neuropilin-1 receptor and triggers nanodrugs penetration into tumor tissues. Ultimately, paclitaxel nanodrugs induce tumor cell cycle arrest and sequential apoptosis.

**Figure 5 nanomaterials-13-01721-f005:**
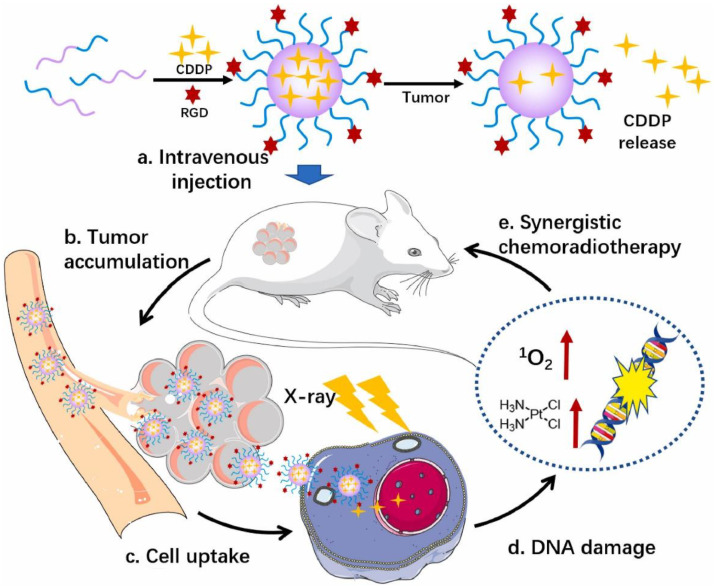
Platinum-based nanoparticles (PtNPs) were functionalized with RGD ligands to selectively target tumor cells and enhance DNA fragmentation [76]. Taking advantage of the targeting performance of RGD, the RPNs can effectively bind to the RGD receptor distributed in the tumor cell membrane to improve drug delivery efficiency and enhance synergistic chemoradiotherapy.

**Figure 6 nanomaterials-13-01721-f006:**
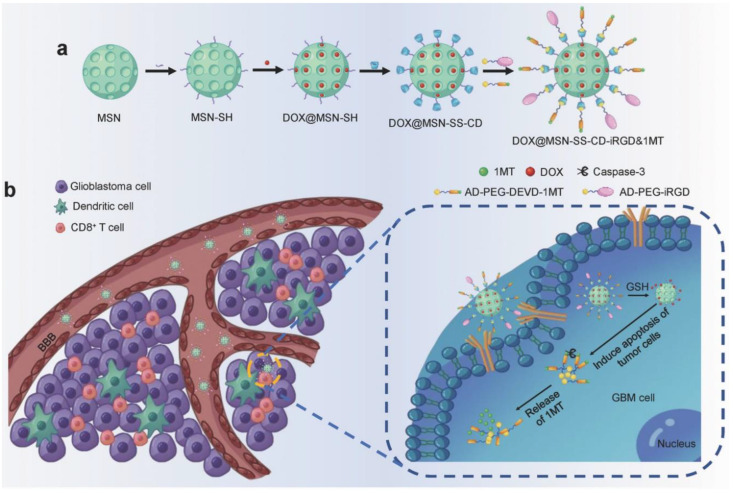
Integrin-targeting DOX@MSN-SS-iRGD&1MT nanoparticles were synthesized to upregulate cytotoxic CD8^+^ T cells for tumor immunotherapy therapy [94]. (**a**) Schematic synthesis of DOX@MSN-SS-iRGD&1MT. (**b**) Schematic illustration of DOX@MSN-SS-iRGD&1MT elicited antitumor immunity against glioblastoma. The introduction of iRGD significantly improves the accumulation efficiency of DOX@MSN-SS-iRGD&1MT into tumors and the cytotoxic CD8^+^ T cell recruitment in tumors.

**Figure 7 nanomaterials-13-01721-f007:**
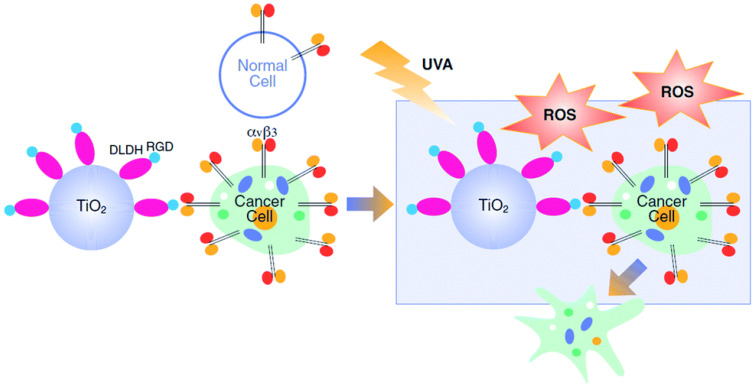
TiO_2_ nanoparticles were modified with RGD groups to promote tumor cell uptake of photosensitizer [105]. Resultantly, the PDT antitumor effect was enhanced with reduced drug resistance.

**Figure 8 nanomaterials-13-01721-f008:**
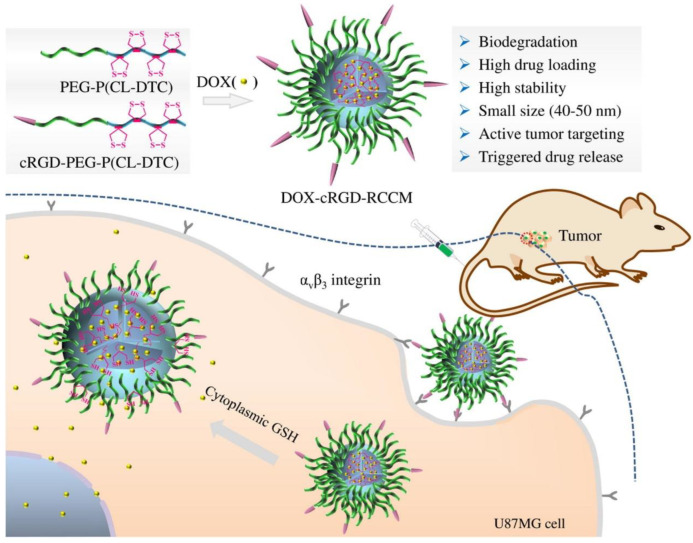
RGD cross-linked nanomicelles showed a long circulation time and efficient glioma accumulation and retention [156].

**Figure 9 nanomaterials-13-01721-f009:**
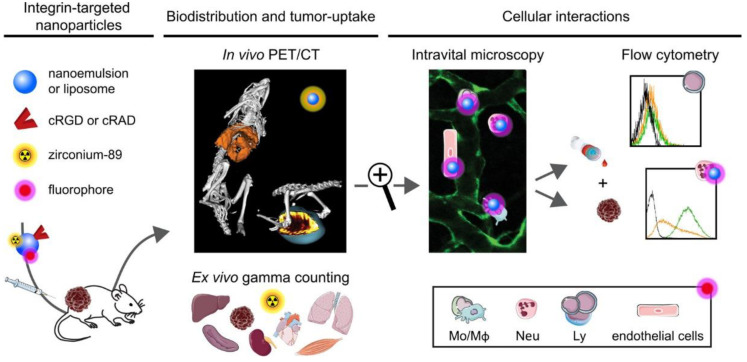
cRGD and non-targeted control polypeptide cRAD were modified with nanoparticles to explore the mechanism of action of targeted drugs in vivo [163]. αvβ3-integrin-targeting nanodrugs were delivered to tumor site via phagocyte hitchhiking.

## Data Availability

In this review manuscript, no new data were created.

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
