# Peer review of "The Role of Integrins for Mediating Nanodrugs to Improve Performance in Tumor Diagnosis and Treatment"

_nanomaterials, 2023, doi:10.3390/nano13111721_

Round 1
Reviewer 1 Report
This is a quite comprehensive review of the role of integrins and nanodrugs as potential therapeutic modalities in cancer and is of potential value. Unfortunately the quality of the written English is low and not at publication quality. The entire article needs to be examined and rewritten by a native English speaker or service. Below I have listed lines where the English is clearly problematic and requires correction for clarity – this may not be an exhaustive list. Secondly, this review manuscript makes extensive use of citing pre-existing reviews. Wherever possible, to make key points it would be preferable to cite one or two key primary data papers to make the points that need to be made. Thirdly, some terminology needs further explanation for non-expert readers.
Lines containing sentences or words that need attention for English quality, often for lack of plural ‘s’:
Line 2 – title – Integrins
15 – faith?
16, 18, 19, 27
34, 38 (attach to ‘interactions’), serve
40
46 – what is meant by ‘modifiers’
47
57 – what is meant by ‘broad’ intracellular domain?
81 – tightly and closely not both needed
85 – signalling molecules
94, 96
Figure 3 – it is not clear to me how this shows outside-in signalling. All it shows is altered integrin conformation
114-123 – as an example, many reviews are cited in this section
124
Reference 62 has no authors
176-177 – grammar?
178 – what is meant by ‘with avb3’?
191
197 – scientific terminology for ‘wrapping radionuclides around’ nanoparticles?
208 – define GTe
213 – define DOX@MSN-SS-iRGD&1MT
223-227 – is more or less accumulation of photosensitiser needed? What is meant by ‘improving photosensitizer and chemotherapeutics to tumor’
236 – ‘nanopdynamical’?
237 – ‘can highly selective to’?
249 – more or less side effects?
255 – could see fluorescence?
257 – a target molecular
281
284 – what are dendritic macromolecules?
295 – what is meant by ‘had an obvious imaging effect’?
309 – define ‘magnetosome’
321 – ‘anti-receptors’?
330 – define ‘opsonicity and clarify sentence
334, 340 – define protein crown
345 – what is DOX-CrGD-RCCMS? This sentence needs to be made more accessible to non-experts
389 – where do immune cells internalise nanomaterials and how do they deliver them to tumours?
392-3 – Sentence needs correction. Is microbe the appropriate word here?
English is not at the level required for publication - see above.
Author Response
Thank you very much for your careful work. We are grateful for your thoughtful suggestions that greatly improved the manuscript. Based on your suggestions, we have proofread and polished the language in this manuscript carefully. Moreover, we plan to seek language editing services from MDPI before this paper was published. In addition, we also made several improvements to original manuscript including the following aspects:
- More detailed discussion of the biological roles mediated by integrins.
- More in-depth conclusion of αvβ3-involved immunotherapy for tumor treatment.
- The interaction of integrin-tumor matrix in both physiological and pathological conditions was carefully reviewed.
- More in-depth conclusion of the challenges and limits of using integrin-targeting nanodrugs in clinical settings.
- A brief overview of different types of integrins and their difference with αvβ3 in terms of their roles in tumor.
- The correlation between the basic science knowledge of integrin and integrins-targeting nanodrugs was emphasized.
- More premium and noteworthy papers related to the topic were concluded and cited.
- All figure captions were rewritten in greater detail.
The main revised content was shown as red words in the paper. Enclosed please find the revised manuscript.

Reviewer 2 Report
This review focuses on integrin: its function as well as its role in tumor therapy. The authors present an in-depth analysis of the structure and function of integrins, the significance of integrin, especially αvβ3 in antitumor therapy, nanoparticles targeting integrin αvβ3, and physical barriers to integrin-targeting nanodrugs. They also discuss various strategies to enhance the efficacy of integrin-targeting nanodrugs and their potential for future applications in tumor diagnosis and treatment. The review is comprehensive and well-structured. The authors include a lot of recent research progress so the audience can understand current state of integrin-targeting nanodrugs. They present the benefits of using nanoparticles for targeted drug delivery. They also cover different therapeutic strategies, including chemotherapy, radiotherapy, immunotherapy. Overall I support the publication but I think there are some issues that the author could improve:
1. The review could benefit from a more detailed discussion of the challenges and limitations associated with the use of integrin-targeting nanodrugs in clinical settings. For instance, potential side effects, safety concerns, and challenges in translating findings to clinical trials.
2. The authors mention all subtypes of integrins at the beginning, and in the conclusion and perspectives the authors also said that other types of integrins such as ανβ6, α4β1, α2β1, and α5β1, are also gaining interest in the field. It would be valuable to provide a brief overview of these integrins and their difference with αvβ3 in terms of their roles in tumor to provide readers with a broader understanding.
3. While the authors provide a thorough introduction to the structure and function of integrins, there appears to be a lack of correlation between this basic science knowledge and the nanodrugs targeting integrin in the later sections of the review. A more in-depth discussion of how the structural aspects of integrins influence their interactions with nanodrugs would be beneficial. In particular, the authors could explore the potential of structure-based drug design in this field.
Overall, this review provides a valuable overview of integrin-targeting nanodrugs for tumor therapy, including their potential benefits. With some improvements in addressing the issues mentioned above, the paper could be accepted for publication.
Author Response
Thank you very much for your careful work. We are grateful for your thoughtful suggestions and valuable comments that greatly improved the manuscript. Based on your suggestions and comments, we have made careful modifications to the manuscript. The major improvements to the original manuscript include the following aspects:
- More in-depth conclusion of the challenges and limits of using integrin-targeting nanodrugs in clinical settings.
- A brief overview of different types of integrins and their difference with αvβ3 in terms of their roles in tumor.
- The correlation between the basic science knowledge of integrin and integrins-targeting nanodrugs was emphasized.
- More detailed discussion of the biological roles mediated by integrins.
- More in-depth conclusion of αvβ3-involved immunotherapy for tumor treatment.
- The interaction of integrin-tumor matrix in both physiological and pathological conditions was carefully reviewed.
- More premium and noteworthy papers related to the topic were concluded and cited.
- All figure captions were rewritten in greater detail and the English language was proofread and polished by a native English researcher.
The modifications were shown as red words in the paper. Enclosed please find the revised manuscript and our point-by-point responses to your suggestions and comments.
Question 1: The review could benefit from a more detailed discussion of the challenges and limitations associated with the use of integrin-targeting nanodrugs in clinical settings. For instance, potential side effects, safety concerns, and challenges in translating findings to clinical trials.
Response: Thank you for your suggestion. We have added more in-depth conclusion of the challenges and limits of using integrin-targeting nanodrugs in clinical settings in the manuscript. Please find Page 21 in the revised work.
Tumor-targeting nanodrugs are an emerging area that has begun to show clinical success, and various nano-based diagnostic and therapeutic drugs are currently undergoing clinical trials (ClinicalTrials.gov). Despite the results to date, there are still many challenges and limitations that need to be resolved. Drug-carriers should effectively deliver cargoes directly to target cells rather than normal organs and tissues. In the meantime, the carriers should possess the ability of biocompatibility and biodegradability, avoiding extra toxicity. Also, these carriers must have long circulation to avoid premature drug release. Integrins possess the ability to specifically interact with target cells to enhance drug delivery efficiency and accuracy. However, integrins are also expressed in normal cells, thus integrin-targeting method exist hidden dangers of off-target effect. This may cause severe side effects in clinical treatments. All in all, we believe that research efforts on integrin-targeting nanodrugs that are used for antitumor therapy will result in highly efficient, safe, curative treatment modalities that meet the prerequisites of personalized medicine and achieve permanent cures without side effects.
Question 2: The authors mention all subtypes of integrins at the beginning, and in the conclusion and perspectives the authors also said that other types of integrins such as ανβ6, α4β1, α2β1, and α5β1, are also gaining interest in the field. It would be valuable to provide a brief overview of these integrins and their difference with αvβ3 in terms of their roles in tumor to provide readers with a broader understanding.
Response: Thank you for your thoughtful suggestion. We have added a brief overview of different types of integrins and their difference with αvβ3 in terms of their roles in tumor in the caption of Figure 1. Please find Page 4 in the revised work.
Figure 1. Classification of integrins family [8]. Integrin heterodimers were composed of different combinations of α and β subunits. In terms of ligand specificity, integrins can be divided into four groups: collagen-binding integrins (α1β1, α2β1, α10β1, and α11β1), RGD-recognizing integrins (α5β1, αVβ1, αVβ3, αVβ5, αVβ6, αVβ8, and αIIbβ3), laminin-binding integrins (α3β1, α6β1, α7β1, and α6β4), and leukocyte integrins (αLβ2, αMβ2, αXβ2, and αDβ2). The β2 integrin subunit (CD18) can pair with one of the four α subunits (αL-CD11a, αM-CD11b, αX-CD11c, and αD-CD11d), forming different leukocyte function-associated antigens.
Question 3: While the authors provide a thorough introduction to the structure and function of integrins, there appears to be a lack of correlation between this basic science knowledge and the nanodrugs targeting integrin in the later sections of the review. A more in-depth discussion of how the structural aspects of integrins influence their interactions with nanodrugs would be beneficial. In particular, the authors could explore the potential of structure-based drug design in this field.
Response: Thank you for your suggestion. We have modified the transition paragraph in Section “Physical barriers to integrin-targeting nanodrug” to emphasize the correlation between the basic science knowledge of integrin and integrins-targeting nanodrugs. Please find Page 16 in the revised work.
Therefore, integrins were often used for specific ligands for targeting drug delivery (targeting tumor cells or ECM). Integrins can be also used to reduce the loss of nanodrugs and increase their half-life and bioavailability in vivo [143,144].

Reviewer 3 Report
Summary:
In this review article, Yu et al., describe the structural characteristics, biological functions, mechanisms of action of integrins in tumor cells and the research progress of the interaction between integrins and tumor microenvironment, as well as the application of integrin-targeted nanomedicine in tumor therapy. Overall, the manuscript deals with a relevant topic in oncobiology, and has scientific merit. However, in my opinion, the authors fail to cite some important papers related to the topic.
Major Concerns:
- Under specific conditions, many integrin ligands, as well as the integrins themselves present structural alterations, which may compromise signaling pathways that are constitutively triggered under physiological conditions. I suggest the authors cite some examples in topic 2.1."The structure of integrins." (PMID: 36429013, PMID: 36566352).
- In topic 2.2 "The function of integrins", the authors discuss about the biological roles mediated by integrins. The topic is well organized, but it is a very rich and complex subject. The authors should explore more in depth on this topic, how integrins can act as transmembrane linkers /integrators, mediating the interactions between the cytoskeleton and the extracellular matrix that are required for cells to grip the matrix (PMID: 36168065, PMID: 23860236).
- It has been very well known, that a potential integrin ligand is the extracellular matrix component fibronectin. However, the authors did not describe about the importance of this interaction (integrin-fibronectin) in both physiological and pathological conditions. I suggest the authors describe a paragraph on this topic, which is relevant to the subject addressed in the manuscript (PMID: 36588107, PMID: 35931112). In addition, there are some articles showing that in the context of therapy against chronic diseases, such as cancer, the pre-treatment of cell monolayers with different matrix components (laminin, fibronectin and collagen), make neoplastic cells more resistant to cytotoxic insults mediated by different classes of chemotherapeutic agents, in an event dependent on integrin-ECM components interaction (PMID: 32118030). Over the last few years, some papers have also demonstrated that chemoresistant cancer cells that overexpress extracellular matrix components are refractory to the toxic effects induced by anti-cancer drugs (PMID: 35816922, PMID: 31987794, PMID: 36882122). Since the biological effects triggered by extracellular matrix components involve the participation of integrins, I suggest the authors address this issue throughout the manuscript, particularly on the topic dealing with cancer therapy.
- The topic "3.3. Immunotherapy" should be reassessed by the authors. In the last 3-4 years, promising studies in the area were developed by many research groups, but they are not mentioned in the manuscript (PMID: 35142593, PMID: 36195696, PMID: 35131862).
Minor Concerns:
- All figure captions must be rewritten in greater detail. The images are nice. However, they bring a lot of information and are poor in content.
- The level of English language is generally good: the manuscript is comprehensible and truly interesting. Nevertheless, proofreading by a native English speaker would be recommended.
As stated above, the manuscript presents scientific relevance, and I only recommend publication after the required corrections.
The level of English language is generally good: the manuscript is comprehensible and truly interesting. Nevertheless, proofreading by a native English speaker would be recommended.
Author Response
Thank you very much for your kind work. We are grateful for your thoughtful suggestions and valuable comments that greatly improved the manuscript. Based on your suggestions and comments, we have made careful modifications to the manuscript. The major improvements to the original manuscript include the following aspects:
- More detailed discussion of the biological roles mediated by integrins.
- More in-depth conclusion of αvβ3-involved immunotherapy for tumor treatment.
- The interaction of integrin-tumor matrix in both physiological and pathological conditions was carefully reviewed.
- More premium and noteworthy papers related to the topic were concluded and cited.
- All figure captions were rewritten in greater detail and the English language was proofread and polished by a native English researcher.
- More in-depth conclusion of the challenges and limits of using integrin-targeting nanodrugs in clinical settings.
- A brief overview of different types of integrins and their difference with αvβ3 in terms of their roles in tumor.
- The correlation between the basic science knowledge of integrin and integrins-targeting nanodrugs was emphasized.
Point-by-point response is added in the paper (red words). Enclosed please find the revised manuscript and our point-by-point responses to your suggestions and comments.
Major Concerns:
- Under specific conditions, many integrin ligands, as well as the integrins themselves present structural alterations, which may compromise signaling pathways that are constitutively triggered under physiological conditions. I suggest the authors cite some examples in topic 2.1."The structure of integrins." (PMID: 36429013, PMID: 36566352).
Response: Thank you for your thoughtful suggestion. We have added these examples to topic 2.1."The structure of integrins." Please find Page 3 in the revised work.
Moreover, many integrin ligands, as well as the integrins themselves present structural alterations under specific conditions, which may compromise signaling pathways that are constitutively triggered under physiological conditions [21,22].
- In topic 2.2 "The function of integrins", the authors discuss about the biological roles mediated by integrins. The topic is well organized, but it is a very rich and complex subject. The authors should explore more in depth on this topic, how integrins can act as transmembrane linkers /integrators, mediating the interactions between the cytoskeleton and the extracellular matrix that are required for cells to grip the matrix (PMID: 36168065, PMID: 23860236).
Response: Thank you for your suggestion. We have conducted a more in-depth discussion of the biological roles mediated by integrins. Please find Page 4 in the revised work.
In this process, the ECM ligand binding would rapidly lead to the conversion of the integrins to the high-affinity extended-open form that permits coupling to the actin cytoskeleton. This bridging to actin supports cell adhesion and the recruitment of additional intracellular binding partners that mechanically reinforce the link and allow subsequent downstream signal transduction [31].
- It has been very well known, that a potential integrin ligand is the extracellular matrix component fibronectin. However, the authors did not describe about the importance of this interaction (integrin-fibronectin) in both physiological and pathological conditions. I suggest the authors describe a paragraph on this topic, which is relevant to the subject addressed in the manuscript (PMID: 36588107, PMID: 35931112). In addition, there are some articles showing that in the context of therapy against chronic diseases, such as cancer, the pre-treatment of cell monolayers with different matrix components (laminin, fibronectin and collagen), make neoplastic cells more resistant to cytotoxic insults mediated by different classes of chemotherapeutic agents, in an event dependent on integrin-ECM components interaction (PMID: 32118030. Over the last few years, some papers have also demonstrated that chemoresistant cancer cells that overexpress extracellular matrix components are refractory to the toxic effects induced by anti-cancer drugs (PMID: 35816922, PMID: 31987794, PMID: 36882122). Since the biological effects triggered by extracellular matrix components involve the participation of integrins, I suggest the authors address this issue throughout the manuscript, particularly on the topic dealing with cancer therapy.)
Response: Thank you for your suggestion. The interaction of ECM components and integrin is essential in tumor progress and chemoresistance. We have carefully reviewed the interaction of integrin-tumor matrix in both physiological and pathological conditions. Please find Page 3 in the revised work. Also, we reorganized the descriptions of the biological effects triggered by ECM components which involve the participation of integrins throughout the paper. Please find in the revised work.
In addition, fibronectin is regarded as a potential integrin ligand, and the extra deposition of fibronectin in tissue was always leading to inflamed diseases [26,27].
- The topic "3.3. Immunotherapy" should be reassessed by the authors. In the last 3-4 years, promising studies in the area were developed by many research groups, but they are not mentioned in the manuscript (PMID: 35142593, PMID: 36195696, PMID: 35131862).
Response: Thank you for your suggestion. We have added and cited several premium and noteworthy papers to the topic "3.3. Immunotherapy". Please find Page 12 in the revised work.
In recent research, an integrin-facilitated lysosomal degradation molecular was designed by a cyclic RGD peptide connecting with PD-L1 [95]. This method allowed both extracellular and cell membrane proteins using bifunctional compounds to facilitate targeted PD-L1 degradation to enhance antitumor immunotherapy. In other researches, inhibition of integrin αvβ6 and αvβ3 also showed a remarkable antitumor effect in different tumors [96–98].
Minor Concerns:
- All figure captions must be rewritten in greater detail. The images are nice. However, they bring a lot of information and are poor in content.
Response: Thank you for your careful work. We have added detailed information and explanations to the figure captions.
- The level of English language is generally good: the manuscript is comprehensible and truly interesting. Nevertheless, proofreading by a native English speaker would be recommended.
Response: Thank you for your suggestion. We plan to seek language editing services from MDPI before this paper was published.

Round 2
Reviewer 1 Report
I am happy to accept the paper following these modifications
Some minor English editing would still be helpful
Reviewer 3 Report
I thank the authors for clarifying my questions. In this new version, all my comments have been properly addressed, and in my opinion the manuscript is suitable for publication.